# Peer review of "Technical Note: Inexpensive modification of Exetainers for the reliable storage of trace-level hydrogen and carbon monoxide gas samples"

_Biogeosciences, 2020_

## Referee Comment (RC1) · Anonymous Referee #1 · 15 Oct 2020

I) General comments

This technical note conveys a solution for field surveys of trace gas fluxes in remote locations requiring the collection of discrete gas samples that are stored for subsequent laboratory analyses. For $H_2$ and CO in particular, the storage of small volume gas samples in glass vials is impeded by $H_2$ and CO emissions from butyl rubber septum fitted to caps.

In a first series of experiments, the authors have carefully tested $H_2$, CO and $CH_4$

emissions from different materials and conditioning protocols. Replacement of conventional butyl rubber septum by silicone plug was proven the most efficient approach to reduce background contamination of $H_2$ and CO. A second experiment has been undertaken to demonstrate performance for long-term (92 days) storage of gas mixtures in modified vials. Stored gas diffusion through silicone was substantially reduced by replacing septum of screwed caps by a stainless-steel bolt and gasket.

Experiments were well conceived, including relevant controls and adequate number of repetitions.

II) Specific comments

- Comparison of $H_2$, CO and $CH_4$ emission rates reported in Figure 2 should be supported by statistical analyses.

- Slope integrating concentration times series in vials presented in figure S1 should be accompanied with standard error to explicitly show variability of reduction or enhancement of trace gas concentration during long-term storage.

- I wonder whether stainless-steel should be replaced with nylon bolt in applications involving survey of marine environments (sea brines cause $H_2$ emissions originating from metal corrosion).

III) Technical corrections

- L91: References are missing.

- L221-223: No data is available to support the statement – better to remove the sentence.

---

## Referee Comment (RC2) · Anonymous Referee #2 · 10 Nov 2020

**Review of "Technical Note: Inexpensive modification of Exetainers for the reliable storage of trace-level hydrogen and carbon monoxide gas samples" by Nauer et al.**

This is a valuable piece of work, well suited for a technical report. The authors produce and test a modified version of commercial Exetainer, that will be useful for many scientists taking gas samples in the field.

$H_2$ and CO are sometimes difficult to preserve in gas samples stored in common containers. Two main processes can modify the mole fraction of $H_2$ and CO: emission from materials in contact with gas (e.g. container walls or septa), and diffusion through container wall or seal. This paper presents a modification of commercial Exetainers in which both these processes are minimized, resulting in an improved gas stability performance.

The paper is well written and to the point. I have only few minor comments as listed below.

**General comments**

- from line 36: The authors compare the convenience of large glass flasks with the small glass vials, but we should be aware that these are used by partly different communities with different requirements. The (1-L and larger) glass flasks are widely used in the atmospheric science community (e.g. NOAA), where often a large air sample is needed. The stability requirements are also much stricter - there, a change in the mole fraction of e.g .CH4 of 2 ppb (0.1%) over several months is already not acceptable (see for examples the WMO compatibility goals - the sample stability should fit well within these limits) (Table 1 in WMO, 2018). The modified Exetainers are useful in situations where signals are large thus precision requirements are more relaxed. Stating this more clearly would be useful.

- some materials emit CO under light. How were the samples stored, in light or dark? Please specify in the method section.

- a short discussion of possible phenomena, and on why these materials were chosen (SS to minimize diffusion through the cap, silicone to minimize the emissions ) may be useful for other scientists trying to make similar experiments for other containers or other gases.

**Specific comments**

- line 23: compared to many other gases in atmosphere, CO and $H_2$ are actually not "highly reactive", as they have lifetimes of several months and 2 years resp. I suggest removing these words.

- lines 36 - 39: glass flask are widely used for atmospheric samples for mole fraction measurements as well, see also general comment

- line 63: the materials were washed and treated, which I assume passivate the surface, but then they were cut into pieces. Does this not counteract the passivation, since it exposes fresh emission surfaces?

- line 80: consider adding the info that the silicone purpose is to keep emissions inside container low

- lines 79-81: consider stating that the silicone and oring were chosen as the best options based on the tests at 2.1? Also, mention the type of oring, and whether it was tested in the previous experiment

- line 92: why did the authors use bolts, and not e.g. a simple round piece of stainless steel?

- lines 182 – 183: unclear, the 0.2 ppm increase in CO cannot be equal to the contamination with a small amount of ambient air as introduced by a needle, since the ambient air is normally around 0.1 to 0.2 ppm.

- lines 188 – 189: the indication of an underlying zero order reaction is interesting, maybe important enough to mention in the abstract? Also, such a zero order (degradation) reaction may be temperature and light dependent – does this suggest that exetainers stored in cold and dark will be more stable?

**Technical comments**

- line 56: 2.2 should be 2.1

- line 91: reference(s) missing

- "concentration" usually refers to mass/volume. The units "ppm", "ppb" normally mean mol/mol (or volume/volume), thus refer to mole fractions or mixing ratios.

- Table 1, caption: part of the text missing?

- Fig. 2: I suggest indicating in the figure caption that the y-axes are different

- line 151: "0.18 ppb" should be "0.18 ppm"

- Figure 4, caption: I think the figure does not show the relative differences (rel dif would be (stored – fresh) / fresh), but the absolute values of fresh and stored gas.

- Supplement figure: I think the "fresh" and stored" are reversed, the stored gas is the one changing.

**References**

NOAA Cooperative Air Sampling Network, https://www.esrl.noaa.gov/gmd/ccgg/flask.html

WMO, 2018: https://library.wmo.int/index.php?lvl=notice_display&id=20698

---

## Author Comment (AC1) · 24 Nov 2020

**The authors would like to thank both reviewers for their expert assessment of our manuscript. We have now duly addressed all comments and suggestions to the best of our knowledge. Please find our response to RC1 from Anonymous Referee 1 below in bold, with page and line numbers referring to the revised manuscript.**

RC1, Anonymous Referee 1

I) General comments

This technical note conveys a solution for field surveys of trace gas fluxes in remote locations requiring the collection of discrete gas samples that are stored for subsequent laboratory analyses. For H2 and CO in particular, the storage of small volume gas samples in glass vials is impeded by H2 and CO emissions from butyl rubber septum fitted to caps.

In a first series of experiments, the authors have carefully tested H2, CO and CH4 emissions from different materials and conditioning protocols. Replacement of conventional butyl rubber septum by silicone plug was proven the most efficient approach to reduce background contamination of H2 and CO. A second experiment has been undertaken to demonstrate performance for long-term (92 days) storage of gas mixtures in modified vials. Stored gas diffusion through silicone was substantially reduced by replacing septum of screwed caps by a stainless-steel bolt and gasket.

Experiments were well conceived, including relevant controls and adequate number of repetitions.

**We thank the reviewer for the positive general assessment of our work. We agree that among the many potential applications, modified Exetainers may be particularly useful for measuring trace-gas fluxes in remote locations.**

II) Specific comments

- Comparison of H2, CO and CH4 emission rates reported in Figure 2 should be supported by statistical analyses.

**Thank you for this helpful suggestion. We have conducted a linear regression analysis to compare the slopes of each gas and treatment to the control and reference gas. Results are summarised in a new Table S1 in the supporting information.**

- Slope integrating concentration times series in vials presented in figure S1 should be accompanied with standard error to explicitly show variability of reduction or enhancement of trace gas concentration during long-term storage.

**We agree with the reviewer. The respective standard errors of the regression slope have now been added to Fig. S1.**

- I wonder whether stainless-steel should be replaced with nylon bolt in applications involving survey of marine environments (sea brines cause H2 emissions originating from metal corrosion).

**This is an excellent suggestion, which we have incorporated in the revised manuscript on p11 l236-239. In our specific application of SEs in marine environments, stainless steel was suitable as gas samples could be kept dry and separate from water samples, but this may not always be possible.**

III) Technical corrections

- L91: References are missing.

**Thank you for pointing this out, the missing references have now been included (p4, l98).**

- L221-223: No data is available to support the statement – better to remove the sentence.

**We agree with the reviewer and have removed the statement in question.**

---

## Author Comment (AC2) · 24 Nov 2020

**The authors would like to thank both reviewers for their expert assessment of our manuscript. We have now duly addressed all comments and suggestions to the best of our knowledge. Please find our response to RC2 from Anonymous Referee 2 below in bold, with page and line numbers referring to the revised manuscript.**

RC2, Anonymous Referee 2

[Figure]

This is a valuable piece of work, well suited for a technical report. The authors produce and test a modified version of commercial Exetainer, that will be useful for many scientists taking gas samples in the field.

H2 and CO are sometimes difficult to preserve in gas samples stored in common containers. Two main processes can modify the mole fraction of H2 and CO: emission from materials in contact with gas (e.g. container walls or septa), and diffusion through container wall or seal. This paper presents a modification of commercial Exetainers in which both these processes are minimized, resulting in an improved gas stability performance.

The paper is well written and to the point. I have only few minor comments as listed below.

**We thank the reviewer for the positive assessment, and are grateful for the detailed and thorough comments listed below, which will greatly improve the manuscript.**

General comments

- from line 36: The authors compare the convenience of large glass flasks with the small glass vials, but we should be aware that these are used by partly different communities with different requirements. The (1-L and larger) glass flasks are widely used in the atmospheric science community (e.g. NOAA), where often a large air sample is needed. The stability requirements are also much stricter - there, a change in the mole fraction of e.g .CH4 of 2 ppb (0.1%) over several months is already not acceptable (see for examples the WMO compatibility goals - the sample stability should fit well within these limits) (Table 1 in WMO, 2018). The modified Exetainers are useful in situations where signals are large thus precision requirements are more relaxed. Stating this more clearly would be useful.

We thank the reviewer for highlighting these important points, which we have now included in the revised manuscript on p2 l38-48:

"...**applications with strict stability requirements of less than a few ppb deviation after many months of storage, e.g. for cooperative atmospheric trace-gas monitoring (https://www.esrl.noaa.gov/gmd/ccgg/flask.html; last accessed 18/11/2020), or sampling campaigns involving...**"

"... **Although these systems represent the 'gold standard' for gas-sample storage, they have two major disadvantages: ...**"

"**In applications where signals are large and thus precision criteria less stringent, e.g. for measuring environmental gas fluxes, glass vials of a few mL volume closed with butyl rubber septa** (...) **are widely used containers to store trace-gas samples.**"

**We fully agree that for certain applications such as atmospheric monitoring there are much stricter requirements which SEs cannot meet. Our SEs are meant to complement, rather than compete against, established and proven gas-sample storage systems customary in certain research fields with stricter criteria.**

- some materials emit CO under light. How were the samples stored, in light or dark? Please specify in the method section.

**All samples were stored in the dark between measurements, and the respective method sections have been amended.**

- a short discussion of possible phenomena, and on why these materials were chosen (SS to minimize diffusion through the cap, silicone to minimize the emissions ) may be useful for other scientists trying to make similar experiments for other containers or other gases.

**We appreciate the reviewers suggestion for an additional discussion paragraph on "possible phenomena" (we assume this means possible degradation mecha-**

nisms of the various rubber materials), and the choice of materials. However, we do not have sufficient data on the specific composition of the rubbers to speculate on potential degradation pathways leading to H2 or CO emissions. We also think this would extend far beyond the purpose of this technical note.

We have now better emphasised the reasoning behind the choice of materials for the SEs throughout the manuscript (see below), and provided suggestions for some alternative materials. We hope this will satisfy what we believe was the essence of the reviewer's request, to provide better guidance for scientists wanting to prepare and adapt SEs for their own use.

p1 l17-18: "...with a silicone plug to minimise contamination, and sealing them with a stainless steel bolt and O-ring as secondary diffusion barrier for long-term storage."

p10 l194-197: "In addition, our selection of silicone sealants was based on local availability and thus rather limited. Although all three products performed reasonably well, there may be better or worse among the multitude of commercially available silicone-based sealants. We thus strongly recommend to test some locally available products for gas release before preparing SEs."

p11 l194-197: "...stainless steel bolt and O-ring as secondary diffusion barrier provided stable storage..."

Specific comments

- line 23: compared to many other gases in atmosphere, CO and H2 are actually not "highly reactive", as they have lifetimes of several months and 2 years resp. I suggest removing these words.

**We agree with the reviewer, the respective words have been removed.**

- lines 36 - 39: glass flask are widely used for atmospheric samples for mole fraction measurements as well, see also general comment

**Thank you, this has been amended according to the suggestions above.**

- line 63: the materials were washed and treated, which I assume passivate the surface, but then they were cut into pieces. Does this not counteract the passivation, since it exposes fresh emission surfaces?

**We agree with the reviewer that cutting was not ideal for the treated materials Tex, Tgr and Tbl. This was however a necessary step to be able to use an autosampler and 12 mL Exetainers for repeated, controlled measurements of the same sample. It also appears that the number of cuts was of minor importance compared to the source/brand of septum. For example Tgr and Tex were both treated grey chlorobutyl rubbers, with Tex cut once and Tgr several times, but the latter showed lower H2 and CO release. Given the relatively poor performance of treated butyl stoppers in general, we believe the small uncertainty introduced by the cutting is justifiable.**

- line 80: consider adding the info that the silicone purpose is to keep emissions inside container low

**Thank you for this valuable suggestion, we have now included this information in the following sentence (p4 l86-87):**

**". . .short-term sample handling, but minimises contamination during long-term storage; and ii) . . ."**

- lines 79-81: consider stating that the silicone and oring were chosen as the best options based on the tests at 2.1? Also, mention the type of oring, and whether it was tested in the previous experiment

**Thank you, we have now included a brief reference to the materials test at the beginning of the paragraph (p4 l84):**

**"Based on the materials test, two simple modifications of Exetainers. . ."**

[Figure]

- line 92: why did the authors use bolts, and not e.g. a simple round piece of stainless steel?

**A valid question and likely of general interest; we have thus included this information in the revised manuscript on p4 l101-104:**

**"In principle, any disk of an impermeable and inert material fitting into the Exetainer cap may serve as secondary diffusion barrier. In our case, using commercial M6 bolts was the most inexpensive and most readily available option, with the rounded buttonhead having the additional benefit of reducing the air cavity between bolt and silicone plug."**

- lines 182 – 183: unclear, the 0.2 ppm increase in CO cannot be equal to the contamination with a small amount of ambient air as introduced by a needle, since the ambient air is normally around 0.1 to 0.2 ppm.

**We thank the reviewer for pointing this out. The comparison with contamination from piercing was referring to H2 only. The text has now been amended accordingly (p10 l192): "For H2, this is close to the contamination introduced by . . ."**

- lines 188 – 189: the indication of an underlying zero order reaction is interesting, maybe important enough to mention in the abstract? Also, such a zero order (degradation) reaction may be temperature and light dependent – does this suggest that exetainers stored in cold and dark will be more stable?

**We thank the reviewer for emphasising our observation of a zero-order degradation reaction, which we have now incorporated in a sentence in the abstract (p1 l14-15):**

**"All tested materials showed a near-linear increase in H2 and CO mixing ratios, indicating a zero-order reaction and material degradation as the underlying cause."**

**As for light dependency, we believe gas samples are routinely stored in the dark, and not much improvement could be gained from further tests. Temperature on the other hand may indeed play an important role for potentially minimizing contamination, and samples stored at lower temperatures could be more stable. However, rubber materials could also become brittle or stiff at low temperatures, which could have adverse effects. A formal investigation of temperature effects on sample contamination was beyond the scope of this study, provided that sufficient results were achieved with SEs stored at room temperature. However, we have mentioned this aspect in the discussion (p10 l207-208) as a recommendation for further investigations.**

**"Temperature may also play an important role for any degradation reaction, and thus merits further investigations for potentially reducing contamination."**

Technical comments

- line 56: 2.2 should be 2.1

**Thank you, this has been corrected.**

- line 91: reference(s) missing

**Thank you, the correct references have been added.**

- "concentration" usually refers to mass/volume. The units "ppm", "ppb" normally mean mol/mol (or volume/volume), thus refer to mole fractions or mixing ratios.

**We agree with the reviewer; the terminology has now been adapted throughout the manuscript.**

- Table 1, caption: part of the text missing?

**Many thanks for highlighting this obvious omission. The missing text has now been added to the caption of Table 1:**

"**Table 1: Treatments for testing rubber materials for H2 and CO release. The control treatments consisted of empty Exetainers. The other treatments contained the listed amount of material, excluding the approximately 0.6 g of pre-treated Exetainer septa used for sealing all treatments and controls.**"

- Fig. 2: I suggest indicating in the figure caption that the y-axes are different

**Thank you for this suggestion, which has now been added to the caption of Fig. 2.**

- line 151: "0.18 ppb" should be "0.18 ppm"

**Thank you, this has been corrected.**

- Figure 4, caption: I think the figure does not show the relative differences (rel dif would be (stored - fresh) / fresh), but the absolute values of fresh and stored gas.

**Indeed, the figure shows measured mixing ratios. The caption has not been updated from a previous version showing relative differences. Thank you for highlighting this error, which has now been corrected:**

"**Fig 4: Measured mixing ratio of reference gases (pressurised air and calibration mix) freshly flushed from cylinder, versus stored for up to three months in Exetainers with different seals. Error bars indicate standard error of the mean. The blue fill indicates the difference between freshly flushed and stored gas is within 5 % of fresh gas. Note that CO was not detected in fresh pressurised air, and the mixing ratio is plotted as zero.**"

- Supplement figure: I think the "fresh" and stored" are reversed, the stored gas is the one changing.

**Thank you, this has now been corrected.**